# Life histories determine divergent population trends for fishes under climate warming

Hui-Yu Wang [1✉], Sheng-Feng Shen [2], Ying-Shiuan Chen[1], Yun-Kae Kiang[1] & Mikko Heino [3,4,5]

Most marine fish species express life-history changes across temperature gradients, such as faster growth, earlier maturation, and higher mortality at higher temperature. However, such climate-driven effects on life histories and population dynamics remain unassessed for most fishes. For 332 Indo-Pacific fishes, we show positive effects of temperature on body growth (but with decreasing asymptotic length), reproductive rates (including earlier age-at-maturation), and natural mortality for all species, with the effect strength varying among habitat-related species groups. Reef and demersal fishes are more sensitive to temperature changes than pelagic and bathydemersal fishes. Using a life table, we show that the combined changes of life histories upon increasing temperature tend to facilitate population growth for slow life-history populations, but reduce it for fast life-history ones. Within our data, lower proportions (25–30%) of slow life-history fishes but greater proportions of fast life-history fishes (42–60%) show declined population growth rates under 1 °C warming. Together, these findings suggest prioritizing sustainable management for fast life-history species.

[1] Institute of Oceanography, National Taiwan University, No. 1, Sec. 4, Roosevelt Rd, Taipei 10617, Taiwan. [2] Biodiversity Research Center, Academia Sinica, No. 128, Sec. 2, Academia Rd, Nankang District, Taipei 11529, Taiwan. [3] Department of Biological Sciences, University of Bergen, P.O. Box 7803, 5020 Bergen, Norway. [4] Institute of Marine Research, P.O. Box 1870, Nordnes, 5817 Bergen, Norway. [5] International Institute for Applied Systems Analysis, A-2361 Laxenburg, Austria. ✉email: huiyuwang@ntu.edu.tw

Climate change is projected to increase ocean temperature, along with driving other physical and biogeochemical changes such as acidification and expansion of hypoxic zones, in marine ecosystems worldwide[1,2]. Such changes are expected to strongly affect marine fauna, particularly in warm regions due to the temperature-dependent oxygen constraints related to increased metabolic oxygen demands and reduced dissolved oxygen supplies[3,4]. Consequently, fish populations in warm regions are expected to show poleward range shifts, leading to declines in potential fisheries catch in the subtropical and tropical regions[5,6]. Furthermore, along with the shifts in distribution, warming will likely impact the population demography and abundance of fishes[7,8]. To date, however, such demographic impacts of climate warming on marine fishes remain equivocal, with some species being positively and some being negatively influenced by warming[9]. Importantly, few have investigated the mechanisms mediating differential population responses to climate change.

Temperature can produce a wide range of within-species variation in life-history characteristics (e.g., growth or reproductive traits[10–12]). Warmer environments are typically associated with smaller body size, higher mortality, faster growth and earlier maturation[13–16]. Because life-history traits determine population growth rates and resilience to disturbances[17], such temperature effects must have implications on population resilience under climate forcing[18,19]. Consequently, life-history traits inform the reference points for fisheries species and likelihood of recovery for the overfished populations[20,21]. Temperature effects on life-history changes and population resistance, however, have mainly been evaluated for single traits (but see[22]). For example, the warming-induced body size decreases have been argued to have dramatic reproductive consequences for marine fishes, given the positively allometric relationship between female body mass and reproductive output[23]. Because of trade-offs between life-history traits and that a combination of these traits determine fitness[24], we posit that evaluation of population resilience requires multiple-trait approaches.

To fill the knowledge gap of temperature effects on life histories and population responses, we compile life-history data and temperature indices from publicly available sources for the Indo-Pacific fishes across 55° S to 65° N (Fig. 1), encompassing a wide range of ambient temperatures and life histories (Supplementary Fig. 1). This region includes several major ocean warming hotspots[25,26], but assessments of temperature-related impacts for most fishes are lacking.

Our life-history data comprise 1402 population records, representing 332 species and 83 families from 440 references published in 1958–2017. For 1268 population records of 321 species (on average, four populations per species, range 1–40; see Supplementary Fig. 2), habitat temperature data are available from NOAA World Atlas 2013, allowing us to evaluate within-species life-history variation in relation to temperature.

We derive two temperature variables, sea surface temperature (SST) and bottom temperature (BT), and calculate minimum, mean, maximum, and coefficient of variation of each of these temperature variables for each population using the long-term average ocean temperature profile from NOAA's World Ocean Atlas 2013 (WOA 13, see "Methods"). We evaluate the effect of each temperature metric on each life-history trait, accounting for the phylogenetic relatedness across a geographic range. We compare the temperature effects on life histories and population growth rates among habitat-associated fish groups. Our findings show that, although warming overall accelerates life-history traits, changes in population growth depend on the fast–slow life-history continuum.

## Results

**Temperature accelerates life-history traits within species.** We found that across all species, increasing mean SST corresponded to trait shifts toward "faster" life histories —i.e., faster growth rates (von Bertalanffy growth coefficient $K$), higher natural mortality rates ($M$; i.e., mortality rate in the absence of fishing;[21,27]), smaller asymptotic lengths ($L_\infty$), and earlier maturation ($A_{50}$) (fixed-effect slopes: $K = 0.05$ yr$^{-1}$ °C$^{-1}$, $P = 0.001$; $M = 0.05$ yr$^{-1}$ °C$^{-1}$, $P = 0.001$; $L_\infty = -0.02$ cm °C$^{-1}$, $P < 0.001$; $A_{50} = -0.04$ yr °C$^{-1}$, $P = 0.03$; Fig. 2a–c, f, respectively; see Supplementary Table 1 for results for other temperature metrics). These slopes are in agreement with the temperature effect on the first principal component (PC1) of these life-history traits (Supplementary Table 2) and are consistent with the mean covariance coefficients (Supplementary Table 3) between temperature and each of the traits in a multivariate life-history model that explicitly accounts for taxonomic structure;[28] together, these results confirm that temperature plays an important role in shaping the life-history variation of fish populations. Furthermore, using splines to represent the temperature effects confirms that the effect of temperature for each of these life-history traits is approximately linear for temperature anomalies of about ±5 °C or even larger, indicating that the majority of populations are well within the species' thermal tolerance windows (Supplementary Fig. 3, Supplementary Table 4); otherwise, strong nonlinear relationships, especially for positive anomalies, are expected[29]. Nonetheless, mean SST did not have a significant effect on the

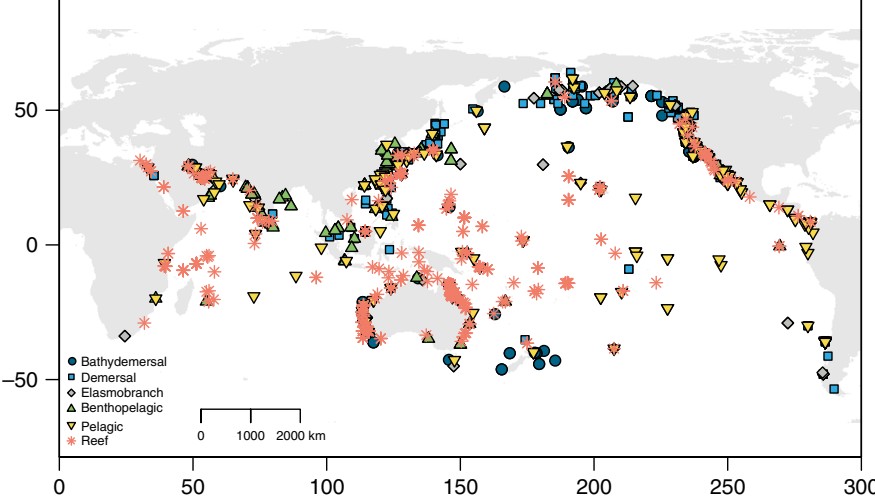

**Fig. 1 Sampling locations of 1402 fish populations in the Indo-Pacific region.** The symbol types denote six groups of fishes.

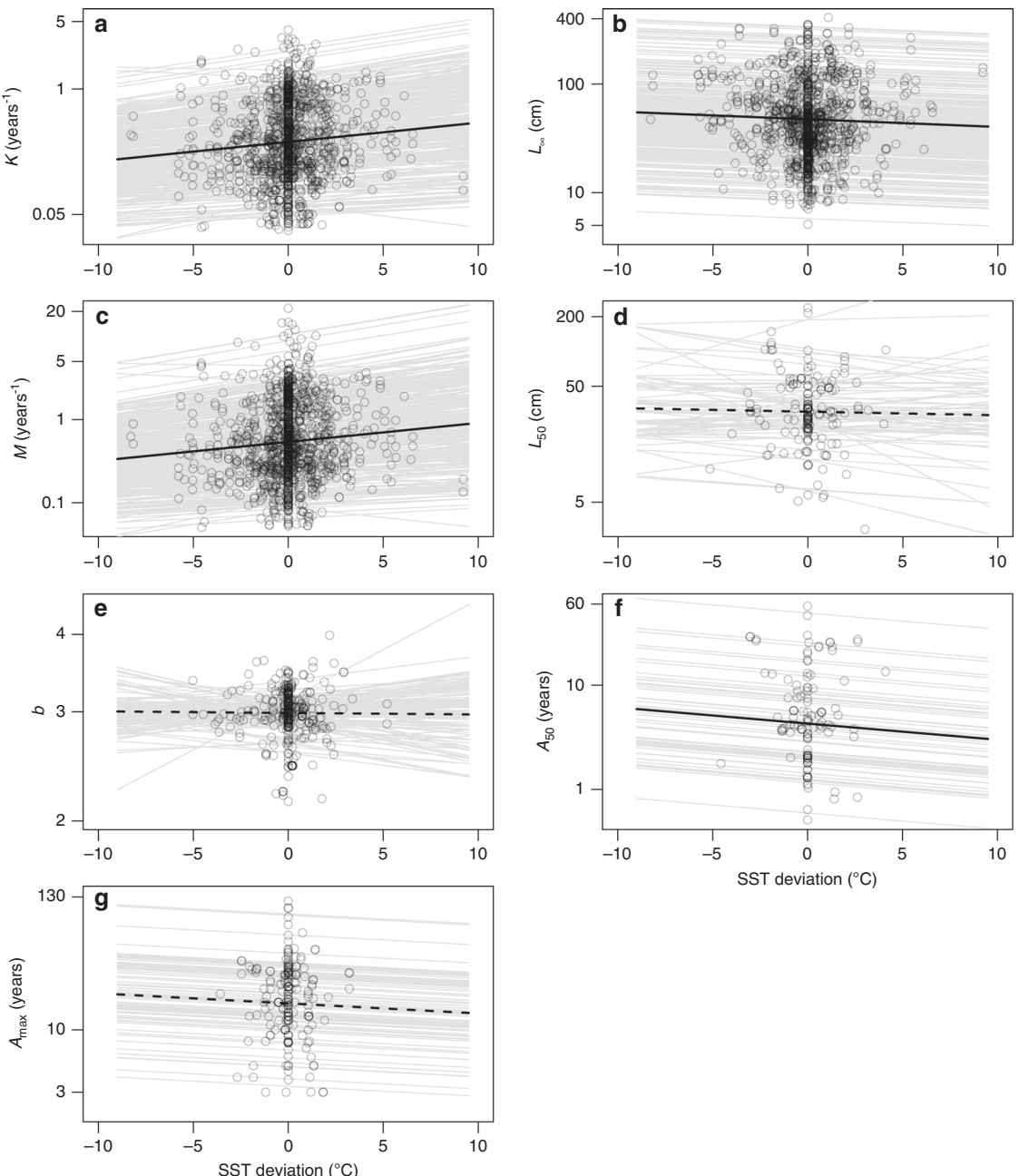

**Fig. 2 Temperature effects on life-history traits for the Indo-Pacific fishes.** The bivariate relationships between temperature deviation (deviation from the species-specific mean sea surface temperature, SST) and each life-history trait are examined using linear mixed-effect models ($K$, $L_\infty$, $M$ (at $A_{50}$), $L_{50}$, $b$, $A_{50}$, and $A_{max}$, **a–g**). Thick black lines represent the tendency lines of the fixed effect of SST deviation (solid and dashed lines, respectively, correspond to significant and non-significant fixed effects), and thin gray lines are the tendency lines for individual species. The vertical lines of data points at mean temperature (anomaly ~0 °C) reflect either a large number of single-population species or reef fishes with narrow ranges of habitat temperatures in our data.

other traits ($b$, $L_{50}$, or $A_{max}$; Fig. 2d–e, g), presumably because fewer observations were available (Supplementary Table 1). Notably, we also found heterogeneous species responses to mean SST for most traits (differences in intercepts and slopes among species in Fig. 2a–g), indicating substantial among species variance (significant species- and family-related variance; Supplementary Table 1).

**Temperature effects vary across the life-history continuum.** Because contrasting environments may cause variation in life-

history responses[30], we compared the temperature-life history correlations among six habitat- and phylogeny-associated functional groups as defined in the FishBase (Supplementary Table 5). These six groups of fishes express temperature-independent differences in their $K$, $L_\infty$, and $M$ values: reef fishes, on average, are characterized by the fastest trait values, whereas the elasmobranch fishes are characterized by the slowest trait values (Fig. 3a–c; Supplementary Fig. 4). These overall differences remain when temperature effects are accounted for (Fig. 3a–c). Moreover, we found consistent signs of temperature effects on the mean responses of $K$, $M$, and $L_\infty$ for all six groups—positive slopes for $K$

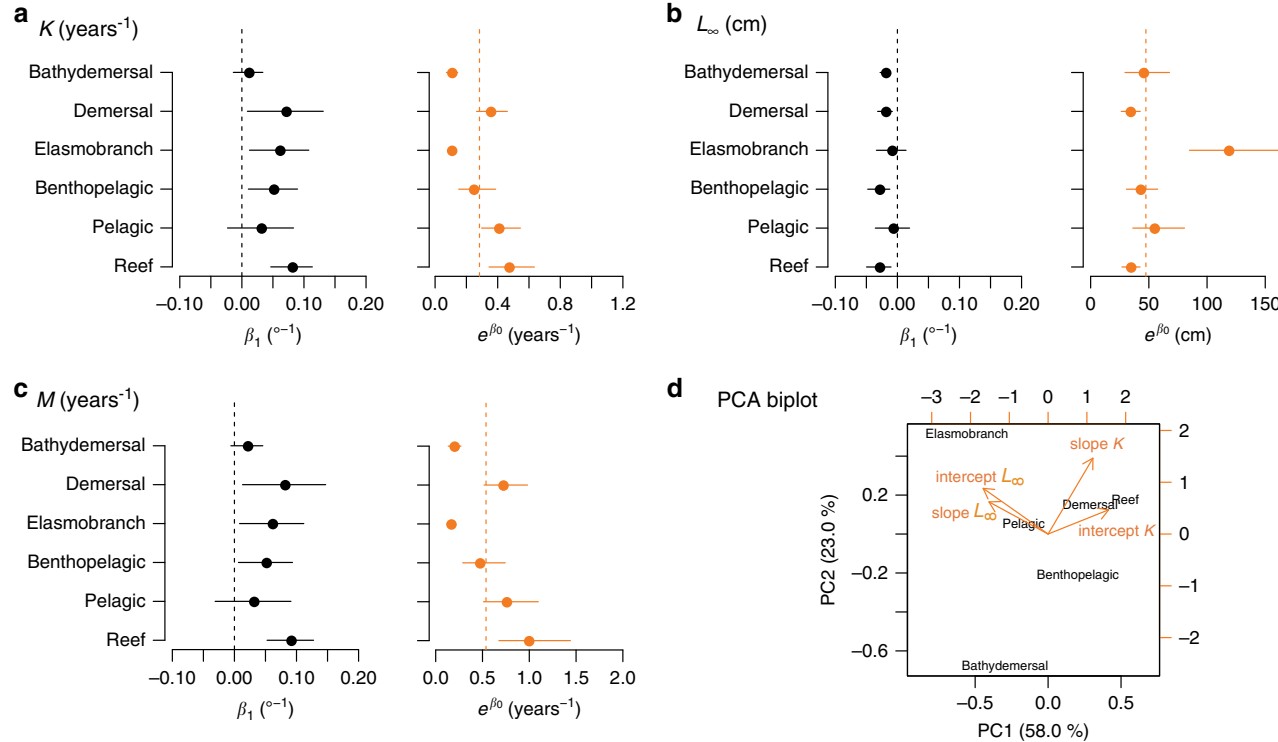

**Fig. 3 Differential temperature effects on life-history traits among six groups of Indo-Pacific fishes.** Temperature effects on three life-history parameters, $K$, $L_\infty$, and $M$ (at $A_{50}$) (**a–c**), are quantified by fitting a linear mixed-effect model for each group with mean-centered SST as the fixed-effect predictor and a ln-transformed life-history parameter as the response. The slopes ($\beta_1$) and back-transformed intercepts ($e^{\beta 0}$) for the six groups of fishes are plotted as the point measures with the 95% confidence intervals as error bars. Vertical dashed lines are reference lines corresponding to $\beta_1 = 0$ (i.e., no temperature sensitivity) and $e^{\beta 0}$ = fixed-effect intercepts pooling all species (mean trait value evaluated at species-specific mean SST). Ordination of the temperature effects on ln $K$ and ln $L_\infty$ for the six groups of fishes based on principal component analysis (PCA; **d**). Black symbols denote the positions of six groups in the reduced dimension space. Orange arrows denote correlation between the temperature effects and the principal components (PC1 and PC2).

and $M$ but negative slopes for $L_\infty$ (Fig. 3a–c; Supplementary Table 6a–c)—even though some of these effects were not significantly different from zero, particularly for the pelagic and bathydemersal fishes (Fig. 3a–c), indicating weak or more variable temperature sensitivities for these groups. Effects of temperature on the other traits for most groups are generally non-significant (Supplementary Table 6d–g). Also, BT exerted weaker effects on life-history traits compared to SST, despite being a relevant temperature measure to demersal or bathydemersal fishes (Supplementary Table 6h–n).

To distinguish differential temperature effects on life histories among groups, we conducted ordination using the principal component analysis (PCA) with variables of the intercepts and slopes of the temperature effects for $K$ and $L_\infty$ ($M$ was omitted because it was derived from $K$ and $L_\infty$) by these six groups. We found that the first PC1 represents a continuum of fast-to-slow life histories (arrows corresponding to the intercepts of $K$ and $L_\infty$ aligned with the $x$-axis and pointing to opposite directions), whereas PC2 was mostly related to temperature sensitivity of $K$ (Fig. 3d). The reef and demersal fishes displayed faster life histories and faster increases of growth and natural mortality under rising temperatures as compared to the elasmobranch and bathydemersal fishes (Fig. 3a, c, d). Furthermore, larger intercept and slope of $L_\infty$ for the elasmobranch fishes indicated their greater $L_\infty$ and smaller reduction in body size with temperature compared to the other fishes (Fig. 3b, d). Such a contrast in $L_\infty$ and rates of reduction in body size may represent a phylogenetic effect on the baseline and temperature-related life-history patterns between elasmobranch and other (i.e., Actinopterygii)

groups. Lastly, the pelagic group at the center of PCA space shows low sensitivity in $K$ and $L_\infty$ to temperature changes (Fig. 3d).

**Temperature rise leads to divergent population growth rates.** Previous research suggests that warming-induced changes in demographic processes (e.g., individual growth, phenology, or recruitment) or range shifts vary among organisms with different life histories or temperature preferences[8,31,32]. Our results corroborate such heterogeneity among a large number of fish species covering all major marine habitat types (Figs. 2, 3). Yet it is difficult to predict how these changes will play out at the population level—facilitating or hampering population growth—when multiple traits are simultaneously responding to temperature change. To understand the temperature-related demographic consequences on population growth, we used a life-table model[33] that allows integrating age-dependent schedules of survival, maturation, and fecundity into lifetime spawning biomass, a measure of reproductive output and an important determinant for the recruitment strength[19,34]. Even though actual recruitment is not possible to predict because of the general lack of data on early-life histories, we can predict the differential effects of temperature change on potential population growth based on the basic reproductive number, $R_0$, which measures the mean per capita offspring production. Consequently, estimating the differential temperature effects on population growth does not require recruitment estimates (but see a recent study providing an assessment of life histories, recruitment, and population growth rate $r$ for about 150 populations[22]). Thus, we parameterized a life-table model with the length-based growth, maturation, and

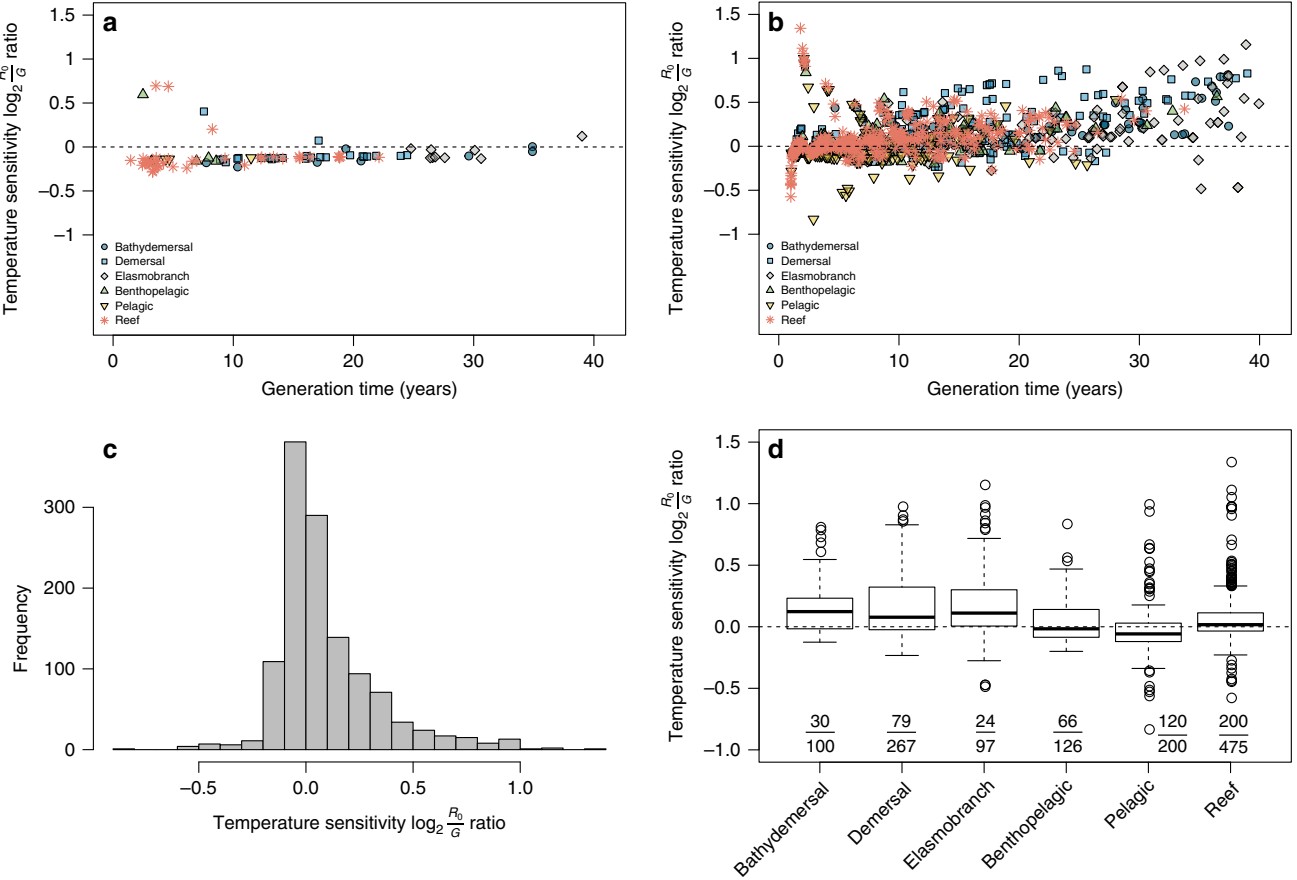

**Fig. 4 Life-table prediction of temperature sensitivity for six groups of Indo-Pacific fishes.** Temperature sensitivity is quantified as the $\log_2$ ratios of annual population growth for 1 °C temperature increase and for baseline temperature ($\log_2 \frac{R_0}{G}$ ratios). Relationships between the temperature sensitivity and generation time are shown for 100 fish populations with empirical body growth and maturation data (**a**) and 1265 populations with empirical body growth data and model-derived age-at-maturation data (**b**). A histogram of the $\log_2 \frac{R_0}{G}$ ratios ($n = 1265$ populations) (**c**). Boxplots of $\log_2 \frac{R_0}{G}$ ratios with proportions of $\log_2 \frac{R_0}{G}$ ratios < 0 by groups ($n = 1265$ populations) (**d**). The lower bounds, centers, and upper bounds of the boxes, respectively, correspond to the 25th, 50th, and 75th percentile of data values within each group. The whiskers extend to the most extreme data point no more than 1.5 times the interquartile range of data within each group.

survivorship functions based on available population-specific life-history data. We then expressed temperature sensitivity as $\log_2$ ratio of annual population growth rates ($\frac{R_0}{G}$, where $G$ is generation time) under the increased (by 1 °C) vs. empirically observed baseline mean temperature (see "Methods"). For the 100 populations with available data of $K$, $L_\infty$, and $A_{50}$, we found negative temperature sensitivity ($\log_2 \frac{R_0}{G}$ ratio < 0: decreased population growth rates under warmer temperature) for most populations except a few short-generation ones (Fig. 4a). We then evaluated $\log_2 \frac{R_0}{G}$ ratios for 1265 populations (Fig. 4b) with empirical $K$ and $L_\infty$ but model-derived $A_{50}$ estimates (see "Methods"). These suggest high variability in temperature sensitivities, with a pre-dominance of positive sensitivity. Negative temperature sensitivity tended to occur for the populations with shorter generation times (Fig. 4b); in total, about 41% ($n = 519$) of our populations showed negative temperature sensitivity (Fig. 4c). Among species groups, the pelagic fishes contained the highest proportion of the populations with negative sensitivity (60%, Fig. 4d). In contrast, high proportions of the populations were predicted to increase population growth rates for the other groups, particularly for the long-generation time populations (Fig. 4b, d). These results suggest that rising temperature will cause uneven impacts on sustainability of fishes in the Indo-Pacific region, with generally

positive impacts on fish populations with slow life histories but negative impacts on fish populations with fast life histories.

## Discussion

Our large-scale within- and among-species comparative study suggests that the warming trend in the Indo-Pacific region[35] will drive increases in the growth coefficient ($K$) and natural mortality ($M$) and decreases in asymptotic length ($L_\infty$) and age-at-50% maturation ($A_{50}$) for fishes. Such "acceleration" of life histories, however, can lead to differential population responses, given species' positions in the fast–slow life-history continuum. Specifically, the groups that are most likely to benefit from warming—elasmobranch, demersal, and bathydemersal fishes—are characterized by slow life histories (relatively long-generation times; Supplementary Fig. 5), while the groups where the overall effect is ambiguous (benthopelagic and reef fishes) or negative (pelagic fishes) tend to have fast life histories. These differences arise from differences in both sensitivity of life-history traits to warming and in life-history traits themselves; i.e., slow life-history species benefit more in terms of reduced mortality and increased fecundity at early ages than they lose later in the life, while fast ones do not gain the early-life benefits but still pay the later costs (Supplementary Fig. 6). Such an effect of inter-specific life-history

variation mediating differential temperature effects on population growth offers new insight into differential population responses to warming temperatures[7–9]. Moreover, our results highlight that assessing population responses to changing temperature requires holistic understanding of life histories, rather than changes of a single trait.

Our finding that climate warming will benefit the slow life-history species but harm fast life-history species is partially corroborated by previous studies. For example, a previous study suggests that fish populations with fast life histories have lower sustainability when experiencing long-term overfishing[36], consistent with our model projection. Small-sized reef fish in Australia have declined in size, while large-sized ones have grown larger[32]. Nonetheless, an inter-specific study of tunas found higher population sustainability for the tropical species with faster life histories compared to temperate ones[15]. The prediction by our model that warming will induce significant declines in population growth rates for many pelagic fishes cannot yet, to our knowledge, be verified with available data. Thus, we urge future studies to continue investigating the role of life histories in determining population resilience under warming and verifying synergies between life-history effects and other factors[9].

We have predicted population growth responses to a warming by 1 °C, a relatively modest degree of warming that is already exceeded in many places. Of course, a population that is able to move poleward or deeper might experience a lesser degree of warming than that observed for a fixed position. Nevertheless, our predictions are best interpreted as proxies of temperature sensitivities, rather than actual responses of specific populations. Similarly, these predictions are subject to uncertainties that reflect the many unknowns in the underlying population processes, yet they are arguably the best predictions that can be derived in the generally data-poor situations that apply to all but a few commercially important fishes.

Our study advances existing knowledge on climate effects on fish populations in several ways. First, our multi-species analysis of temperature effects on life-history traits generalizes previous studies of temperature-life-history patterns for single species (e.g., Atlantic cod *Gadus morhua*,[13]; little skate *Leucoraja erinacea*,[37]) or for particular species groups (e.g., tunas,[15], reef fish,[32]). Also, exploring temperature effects on multiple life-history traits broadens previous studies that primarily focused on the temperature-size relationship only[16] (but see[28]). More importantly, our model highlights the role of natural mortality and fecundity as suggested in previous work[19,21], in determining population growth rates. Lastly, the existing assessments of climate impacts on fisheries catch have assumed a constant warming effect on life histories for all fishes[38,39]. Regarding such assessments, our study provides critical data to account for differential life-history responses under warming, increasing accuracy in the projection of warming impacts on fisheries production.

With rising human demand for seafood[40], it is important to understand and mitigate the effects of climate change on fish populations[41]. Most present studies suggest that the negative effect of temperature on body size could lead to adverse size-related changes, such as disruption in trophic interactions or reduced reproductive capacity and productivity for fish populations[23,38,42]. In contrast, our study demonstrates that rising temperature promotes faster life histories, with positive or negative influences on population growth rates, depending on populations' positions on the slow–fast continuum. Given that recent research has found a mix of positive, negative, and neutral effects of historical climate warming on population growth rate[9] —differing from the forecast results of most existing models— more studies simultaneously evaluating the positive and negative

temperature-related effects and incorporating life-history evolution to understand population resilience under changing climate are urgently needed.

## Methods

**Compilation of population life-history data**. We compiled population-specific life-history data for Indo-Pacific fishes from primary literature using a combination of systematic and opportunistic searches. We obtained references primarily from Google Scholar and the web page of Fisheries and Oceans Canada (DFO, http://www.isdm-gdsi.gc.ca/csas-sccs/applications/publications/index-eng.asp), using a combination of species name and the following keywords: Pacific, temperature effects, von Bertalanffy growth, age at maturation, or natural mortality. Species names for 107 common teleosts in the Indo-Pacific Ocean were obtained from the global-capture production database of the Food and Agricultural Organization (FAO, ISSCAAP groups 31-37; available at: http://www.fao.org/fishery/statistics/global-capture-production/query/en). Although similar data are available in *FishBase*, a global and freely available database on fish, we opted not to use data from *FishBase* because temperature data associated with population records are often missing and difficult to restore due to lacking spatial coordinates in *FishBase*.

In total, we screened around 8000 references published in 1958–2017. We extracted data from 440 references representing peer-reviewed journal papers, stock assessment reports, governmental documents, graduate theses, and conference proceedings. Our data include 332 species, with conspecific life-history data of ≥2 records in 206 species.

We extracted available population data of seven life-history traits related to growth and maturation for each population: i.e., the von Bertalanffy growth coefficient $K$ (yr$^{-1}$) and asymptotic length $L_\infty$ (cm), age and length at 50% maturity ($A_{50}$ (yrs) and $L_{50}$ (cm)), allometric exponent ($b$) of length–weight relationship $W = aL^b$, lifespan ($A_{max}$ (yrs)), and natural mortality (yr$^{-1}$). To make the length traits ($L_\infty$ and $L_{50}$) comparable among populations, we standardized different length measures (e.g., standard length or fork length) into total length using available regression models (Supplementary Table 7). To account for between-sex differences in traits values, we coded life-history data of each population as sex-specific or combined-sexes. When data of single- and combined-sexes were both available, we primarily used combined-sexes data for our analysis. When only sex-specific data were available, we used female data.

In our selected references, the von Bertalanffy growth coefficients are primarily estimated via fitting length-at-age data ($L_t$, where $L$ is length and $t$ is age) with one of these methods: the Ford–Walford method, length-based method, and nonlinear regression fitting. As all these fitting methods involve the same model form $L_t = L_\infty\left(1 - e^{-K(t-t_0)}\right)$ and similar optimization processes (e.g., minimizing residual errors), we consider that the growth coefficients estimated with these methods are generally comparable. However, we excluded the growth coefficient data derived from other model forms, as transforming those coefficients could introduce biases (e.g., Gompertz model or weight-based form of von Bertalanffy model). The population natural mortality data were derived from various models with distinct assumptions (Supplementary Table 8); thus, these natural mortality estimates may not be comparable. To investigate the temperature effects on natural mortality, we re-estimated natural mortality estimates for individual populations using a life-history invariant method (see Methods: *Derivation of M*).

We recorded additional descriptive variables for each population: e.g., ecological group (coded based on habitat types following definitions in *FishBase*, Supplementary Table 5), family, species, latitude, and longitude of sampling site (in decimal degrees; for the studies with multiple sampling sites, the minimum and maximum of latitudes and longitudes of sampling sites are recorded), minimum and maximum sampling depths, and sample size. Latitudes and longitudes of sampling sites were derived from available spatial coordinates, names of sampling ports, or the maps in the references.

Some references were excluded to avoid ambiguity. For example, we excluded references of fish in estuary and artificial habitats (e.g., aquaculture species). To alleviate noise in the datasets, we rejected data lacking clear descriptions of study area (e.g., studies that did not define sampling sites or those that aggregated samples across a very broad range) and those lacking explicit information on fitting procedures for the growth or maturation models (e.g., lacking information on sample size, suitable range of length-at-age data, independent samples, description of the fitting methods, a plot of data with the fitted line, or other means for assessing the fit). Similarly, we excluded studies without descriptions of fitting methods for estimating natural mortality (as suggested by 14). For stock assessment reports and review papers, we scrutinized the summarized data based on original publications and rejected data lacking clear sampling or fitting information. Lastly, we removed duplicated data cited in multiple reports.

**Compilation of population temperature data**. We used available in situ data of mean decadal sea temperature as habitat data. These temperature data were obtained from the World Ocean Atlas 2013 (NOAA Atlas NESDIS 73; 43; hereafter referred to as WOA13; available at: https://www.nodc.noaa.gov/cgi-bin/OC5/woa13). Sources of the WOA13 temperature data include temperature profiles measured by various instruments[43]. The mean decadal temperature profile was calculated by averaging six decadal datasets spanning 1955–2012[43]. For our analysis, we used the mean decadal

temperature profile data in the Indo-Pacific region, with a horizontal resolution of 0.25° and depth segments of 5 m from surface to 100 m and depth segments of 25 −100 m for >100 m (maximum depth = 5500 m)[43]. As rates of temporal changes in the ocean temperature were slow (e.g., 1.48 °C per century; 25), we did not account for the small temperature changes due to differences in time of measurements between the temperature and life-history data.

We matched the mean decadal temperature profile data with spatial coordinates for each of these populations. For populations with single sampling sites, we assumed that their potential habitats centered at the sampling sites and extended for 0.5° in the north, south, east, and west directions. Further, for populations with multiple sampling sites, we assumed that their habitats were bounded by the ranges of latitudes and longitudes of sampling sites. To account for habitat depths, we estimated the minimum and maximum depths of populations based on ranges of sampling depths or description of depths of species habitats in the references. When depth information was unavailable, we used the maximum depth of the species in FishBase as the maximum depth for a population. For each population, we derived two habitat variables: SST and BT. SST is the temperature at 0 m (WOA13 data; 43). BT is the temperature at the maximum depths of populations. Finally, we calculated the minimum, mean, maximum, and coefficients of variation of each of the habitat indices for each population.

**Derivation of natural mortality (M).** Population-specific natural mortality (hereafter referred as $M$) provides insight into population sustainability under fishing or environmental changes (i.e., high natural mortality indicates high degree of sustainability[13,15]). However, because estimates of $M$ from different models are not necessarily comparable, we re-estimated $M$ for each population using the model II of Gislason et al.[14], which builds on well-established empirical relationships[44,45] and has some theoretical backing[46,47]. The equation is:

$$\ln(M) = 0.55 - 1.61 \ln(L) + 1.44 \ln(L_\infty) + \ln(K), \qquad (1)$$

where $M$ is natural mortality and $L$ is the midpoint of length range (cm) of a population, respectively[14]. As the length range data were not available to us, we constrained $L$ to be the length at age-at-50% maturity ($A_{50}$) to estimate $M$, following the invariant relationship among mortality-at-age at maturation, $L_{50}$, $L_\infty$ and $K$[13,48]. Further, because $A_{50}$ data are available for fewer number of populations ($n = 119$, about 8.5% of total populations), we restored the missing values of $A_{50}$ based on a theoretical linear relationship between population $A_{50}$ and $\frac{L_{50}}{K \times L_\infty}$[49]. With the available data of $A_{50}$, $L_{50}$, $L_\infty$, and $K$ for 70 populations, we fitted this linear relationship ($F = 204.6$, df = 1,68, $R^2 = 0.75$, $P < 0.001$; Supplementary Fig. 7):

$$\frac{L_{50}}{K \times L_\infty} = 0.084 + 0.635 \times A_{50}. \qquad (2)$$

Using Eq. (2) and by approximating population $L_{50}$ by $2/3* L_\infty$[48], we restored missing $A_{50}$ data and computed $L$ (i.e., plugging $A_{50}$ data into a von Bertalanffy growth function). Then, with these estimates of population $L$, $L_\infty$, and $K$, we used Eq. (1) to estimate of M for all populations. We excluded 15 populations of 3 species with estimates of M of exceptionally high values ($M > 25$ yr$^{-1}$). Range of $M$'s for the remaining 1387 populations is 0.05–21.8 yr$^{-1}$ (Supplementary Fig. 1). These Gislason $M$ estimates were used in the subsequent regression and life-table modeling analysis.

**Evaluation of the relationships between temperature and life-history traits.** Because life-history variation is nested within phylogenetic levels, we used the linear mixed-effect model (LME) to explore the relationships between each life-history trait and each temperature index, simultaneously accounting for species- and family-related variance as random effects. We used the lme4 and lmetest packages in R (www.r-project.org[50];) to construct these LME models. Given the eight different temperature indices (e.g., four descriptive metrics nested within two temperature variables), we constructed eight LME models for each life-history trait. For each model, a natural-log transformed life-history trait is the response variable, a species mean-centered habitat index is a single fixed-effect variable, and species and family are two random-effect variables. We considered four alternative model structures for random effects: i.e., with either family or species as random intercepts, and with either family or species as random intercepts and slopes.

We evaluated strength of each of fixed- and random-effect variables using the likelihood ratio test[51]. Specifically, we estimated the log likelihood between a pair of full and alternative models (with one less fixed or random variables), deriving the test statistics (i.e., 2 times the difference between log likelihood of two models) and $P$ value. If the two models were not different significantly (e.g., $P > 0.05$), we selected the more parsimonious model. Otherwise, significant difference between these two models (e.g., $P \leq 0.05$) indicates pronounced variation in the response variable due to the additional fixed or random effect in the full model. Furthermore, when neither family nor species accounts for significant variance in the response variable, we reduced the model to a simple linear regression with a habitat index as the sole fixed-effect predictor. We selected the best model structure to depict the relationship between each of the eight habitat indices and a life-history trait.

We observed pronounced intra-specific variability in the empirical $K$ and $L_\infty$ for some of our study species; e.g., ≥3-fold differences in these traits within some species. Because large variability in $K$ and $L_\infty$ within a species may be partially due

to that the estimation of these traits depends on one another, we conducted PCA with these two traits and evaluated the ratios of maximum-to-minimum scores of the first PC1 among species. We found 34 species with the absolute values of the ratios of PC1 scores ≥3, whereas 285 species had |ratios| <3 (Supplementary Fig. 8). To account for such within-species variability in these traits, we compared results of the LMEs with data of all populations vs. those with data excluding the species with the |ratios| ≥3. We included results of the LMEs with the reduced data in Supplementary Table 9.

To evaluate if species groups mediate differential responses to temperature, we examined the effect of mean SST (fixed-effect variable) on each life-history trait for each group using LME models. For each group, we calculated the 95% confidence intervals of the slopes and intercepts of the fixed effects as: $\beta \pm t_{0.025, n-1} \times \text{SE}(\beta)$, where $\beta$ represents an estimate of slope or intercept. We evaluated patterns of the slopes and intercepts among groups using the PCA with the built-in function prcomp in R.

**Evaluation of warming effects on population growth rates.** To evaluate the effects of temperature-induced life-history changes on population growth rates, we compared the annual population growth rates ($\frac{R_0}{G}$, where $G$ is generation time) at 1 °C increase from the mean temperature to that at the mean temperature for each population (i.e., by calculating the $\log_2$ ratio of $\frac{R_0}{G}$ at +1 °C to $\frac{R_0}{G}$ at mean SST, hereafter referred as $\frac{R_0}{G}$ ratios; $\frac{R_0}{G}$ ratios >0 indicate positive warming effects on population growth rates). To derive $\frac{R_0}{G}$, we built an age-structured model (i.e., a life-table[33];), incorporating functions for growth increment (in length and weight), maturity states, fecundity, and survivorship probability. The length-at-age data ($L_t$) were calculated using the von Bertalanffy growth model (Eq. (3)):

$$L_t = L_\infty \left(1 - e^{-K(t-t_0)}\right), \qquad (3)$$

where $t$ denotes age, and $L_\infty$, $K$, and $t_0$ are the von Bertalanffy growth coefficients. Weight-at-age data ($W_t$) were estimated using an allometric function of $L_t$ (Eq. (4)):

$$W_t = \alpha L_t^\beta. \qquad (4)$$

Also, fecundity ($m_t$) is an allometric function of $W_t$ (Eq. (5))[23]:

$$m_t = \delta W_t^\gamma. \qquad (5)$$

Because of lacking data on the length-weight and fecundity-weight relationships for many populations, we assumed constant intercepts and slopes for Eqs. (4) and (5) (i.e., $\alpha = 0.02$, $\beta = 3$, $\gamma = 1.18$, $\delta = 2930$). However, $L_\infty$ and $K$ data are available for most populations in our data ($n = 1,387$). As a result, we used these data to derive length-at-age for each population (assuming $t_0 = 0$). Also, we used the model-derived $A_{50}$ estimates (Supplementary Fig. 7) to account for maturity state for $m_t$ for each population (i.e., $m_t = 0$ for $t < A_{50}$; $m_t = \delta W_t^\gamma$ for $t \geq A_{50}$). Survivorship-at-age data ($l_t$) were estimated from natural mortality-at-age ($M_t$) (Eq. (6)):

$$l_t = \frac{\prod_{i=0}^{i=t-1} e^{-M_i}}{e^{-M_0}}. \qquad (6)$$

We used Eq. (1), substituting $L_t$ for $L$, to derive $M_t$.

We estimated the ratio of lifetime net reproduction ($R_0$) over the generation time ($G$, the average reproductive age; Eq. (7)):

$$\frac{R_0}{G} = \frac{\sum_{t=0}^{t=\max t} l_t m_t}{\frac{\sum_{t=0}^{t=\max t} t l_t m_t}{\sum_{t=0}^{t=\max t} l_t m_t}}. \qquad (7)$$

Because of lacking lifespan data for many populations, we assumed a constant maximum age to be 50 years (max $t$ in Eq. (7)) for all populations.

We incorporated the differential temperature slopes of $K$, $L_\infty$, and $A_{50}$ to calculate the $\frac{R_0}{G}$ ratios for 100 populations with available data of $K$, $L_\infty$, and $A_{50}$. Subsequently, we also evaluated $\frac{R_0}{G}$ ratios for 1265 populations with available data of $K$ and $L_\infty$ and model-derived $A_{50}$ estimates, incorporating differential temperature slopes on each of these traits.

**Reporting summary.** Further information on experimental design is available in the Nature Research Reporting Summary linked to this paper.

## Data availability

The data that support the findings of this study are available from the corresponding author upon reasonable request. DFO database: http://www.isdm-gdsi.gc.ca/csas-sccs/applications/publications/index-eng.asp. Global capture production database. FishBase. NOAA's World Ocean Atlas 2013

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

## Acknowledgements

This research was supported by Ministry of Science & Technology of Taiwan, grants MOST 104-2611-M-002-016, MOST 105-2611-M-002-002 and MOST 105-2811-M-002-068, and by Norwegian Research Council (288037).

## Author contributions

H.-Y.W. conceived the project. Y.-S.C., Y.-K.K., and H.-Y.W. compiled data. S.-F.S., M.H., and H.-Y.W. analyzed data, conducted modeling analysis, and wrote the manuscript. All authors contributed to revisions of the manuscript.

## Competing interests

The authors declare no competing interests.
