## [Peer Review File · Nature Communications]

Peer Review File –

Reviewers' comments first round:

Reviewer #1 (Remarks to the Author):

This study quantifies the relationship between ocean temperature and seven life history traits for 332 fish species occurring in the Pacific and Indian Ocean basins and the expected cumulative impacts of these relationships on population growth rates under a degree of warming. The authors individually regress each life history trait against temperature. They then use life tables to derive and compare the intrinsic rate of population growth at average temperature versus one degree warmer than average by using their individual regressions to predict life history traits at each of those temperatures.

My primary concern is that the authors predict the impact of temperature on each life history trait individually (seven univariate models) rather than in a single multivariate model. In fact, life history traits are correlated to each other in addition to temperature and the univariate approach does not account for this covariance. Thorson et al. (2017) use a multivariate phylogenetic model to predict the relationships between temperature and a highly similar set of life history traits for all fish species using life history data from FishBase. The authors should use this or a similar modeling approach to re-estimate the influence of temperature on life history accounting for correlations between traits and across phylogenetic relatedness. Fortunately, Thorson et al. (2017) provide an R package that allows their model to be refit to a new set of data: <https://github.com/James-Thorson/FishLife>

Thorson, J. T., S. B. Munch, J. M. Cope, and J. Gao. 2017. Predicting life history parameters for all fishes worldwide. *Ecological Applications* 27(8): 2262-2276.

My secondary concern is the extent to which these results can be used to predict cumulative impacts on the intrinsic growth rate on the entire population given the following argument: if populations are shifting distributions to follow their preferred temperatures shouldn't the average temperature remain the same through time and thus no result in a change in population-wide intrinsic growth rate (just changes in latitude-specific growth rates)? To make this argument hold water, the authors will need to present the evidence that as populations shift distributions in response to climate change, the mean temperature will increase or the distribution of temperatures will increase such that the spatial distribution shifts don't equalize impacts on life history.

Related to this concern is the fact that the influence of temperature on life history traits is dome-shaped and not monotonic and that neither the univariate or multivariate approaches account for this. Ultimately, I think it's okay because shifting distributions will prevent mean temperatures from moving too far past the thermal optimum but I think this should be acknowledge in the manuscript. For example, in Figure 2, growth rate can't continue to increase forever under increasing temperatures, it will decrease past some threshold.

My final big concern is the straight vertical line of data points in Figure 2 – what is going on there?

Small comments

25 – what's the connection between reproductive rates and asymptotic rate and age-at-maturation? Should these be separate traits in the list?

27 – delete colon and make separate sentence

27 – species groups appears to be based on habitat associations -- reword to convey this

28 – move this sentence summarizing results after the sentence summarizing methods (the life table sentence)

31 – age-specific

34 – and lower proportions in what groups?

40 – ocean warming isn't contributing to acidification – reword

50 – there's been lots of work on temperature effects of recruitment and on somatic growth rates that is not cited here

51 – Rewrite to say that temperature can produce intraspecific variation in life history

61 – positive allometric

64 – The study question, region, and methods have not been introduced at all. You need a paragraph introducing the purpose of the study and general methods.

69 – this is half the world's oceans – saying that it's where some of the most rapid warming has occurred isn't that interesting or profound. And isn't it not true? Isn't the Northeast US and the southern Ocean the fastest warming?

79 – derived temperature variables from what dataset? At what time of year? Is it matched to the time the life history trait was measured?

88 – I think you should state that this is a life-history invariant method so that the reader knows it was derived from other life history variables

92 – across all species

98 – would be useful to see sample sizes somewhere

104 – why did you define species groups based on habitat affiliation and not some other way of grouping species (trophic level, taxonomic order, functional group, etc).

104 – define groups here

132 – Rewrite the sentence to say that use a life table to "insert what you use life table to do". This will more clearly set up readers expectations for what the purpose the life table is.

144-146 fix grammar of this sentence

150 – where do you show that populations in warmer habitats were more negatively impacted? I don't see this evidence presented.

185 – I think this grossly ignores the Thorson et al. paper that examines the influence of temperature on all of these life history traits simultaneously across all taxa. In general, this work needs to be better acknowledged throughout the paper.

201 - this study doesn't say there has be no overall effect. It says there have been some winners, some losers, and some unaffected, and that the losers outweigh the winners.

342 – how would your life history trait database look differently if you had drawn the life history data from FishBase using the rfishbase R package? Do you have more data? Less data?

362-365 – If these natural mortality estimates aren't comparable, why do you include them together in the same regression?

393-387? What is the temporal resolution matched to the life history data? The mean decadal temperature for that cell? Can you clarify this?

435 – I'm curious about these species. Are the squids? Shrimps?

436 – Plot histograms of life history parameters in supplement?

500-502 – I thought the whole point is that a rise in temperature has different effects on different species? Is this the mean?

Table and Figure comments

All of the figures require lots of cosmetic work to make them up to the standards of a Nature journal.

Fig 2 – what are the straight lines at zero? Color by group and show group fixed effect? What about random effect by group since you have that hypothesis? No p-values shown here. Mark significance. Why no lines for some? Why monotonic and not dome shaped

Fig 3 – Why is intercept interesting? Why not remove the intercept plots and show the slope plots for all of the evaluated life history parameters instead?

Fig 4 – In caption, define what values above and below zero mean. Explain the bimodal shape in panels A and B. Color points by habitat association?

Figure S1 – how is a population defined and delimited?

Reviewer #2 (Remarks to the Author):

This study appears to show that life-history responses to warming differ in relation to phylogeny and habitat with reef and demersal fishes being more sensitive to temperature change than bathydemersal fishes. Interestingly, rising temperature while promoting faster life-history traits, seems to have positive or negative influences on population growth, depending on populations' position on a life-history continuum. The evidence presented that fast life-history populations are more vulnerable to warming compared to slow life-history ones is particularly important in the debate over climate change impacts.

The authors attempt to simultaneously account for multiple traits in predicting the consequences of warming which is an important advance over many published studies that have only considered single traits. The number of species and populations considered is also impressive. The methodological approach taken seems reasonable and appropriate given the type of data available for so many species. The statistical analysis using LME for comparing LH traits and temperature and the PCA seem appropriate and the results are clearly explained.

Considering natural mortality for so many species is a problem to any large comparative study so I understand why the authors have used the Gislason et al. (2010) relationship. While the authors of that study did suggest that temperature could be used to explore some aspects of warming, I am not clear as to why estimation of M for the length at age-at-50% maturity provides a good basis for inter-species comparison.

Although I appreciate why SST and BT were used as proxies I did not understand why only model fits for the relationships between mean SST and each life history trait by six groups of fishes were presented in Table S2, as while SST may be relevant for pelagic fish it won't be for bathydemersal species.

While I think the major claims of the study will need to be explored in greater detail at the species or guild level, I think this study does provide an important comparative advance that will interest climate scientists and fishery managers.

Minor comments

Did the authors really mean to say that fish adapt to temperature increases by displaying faster life-history strategies such higher mortality?

Given that temperature is not the only driver of long term life history traits, compiling life history data from 1958 – 2017 may partially compromise the comparative analysis. I would like to hear the authors view as to whether this is important to the study. Personally, I doubt it will be a major weakness of the study as differences among taxa are generally greater than intra-specific variation. As authors say on line 102 there significant species- and family-related variance was found for most life-history traits.

Reviewer #3 (Remarks to the Author):

This study explores the relationship between water temperature and several life-history characteristics in a large literature derived dataset of over 300 fish species from Indian and Pacific Oceans. To assess how populations of these species are likely to respond to warming the authors apply life-history tables to estimate population growth rates under changing life-history characteristics.

I applaud the authors for compiling this dataset and would strongly urge that upon the publication the dataset is made public with FULL references included (at the moment the dataset I see only has year and author name). However, my understanding is that the authors use Von Bertalanffy growth curve coefficients K and Linf from comparable models (line 361) without assessing the quality of the actual dataset to which the curves were fit. I realise such specific checking would perhaps be too big of a task, however, fitting VB curves is not straightforward, and the procedure is sensitive to the absence of oldest (in case of heavily fished stocks) or youngest (in case of fishery dependent datasets) age classes in the data. My quick look through the data set shows large amounts of intra-specific variation. For example, entry 11 shows K and Linf of *Trichiurus lepturus* to be at 0.41 and 61.5cm, whereas entry 6 for the same species (and not that distant a geographic location) shows 0.116 and 193.8 cm. Is such a 3-fold difference in Linf reasonable? If I filter species that have at least five entries and calculate the ratio between min and max Linf values for each species, I find that 20 out of 81 species have ratios >2 (one species has a ratio of 11). For the coefficient K, half of the ratios are >3. This would correspond to values of, say, 0.05 and 0.15 for the same species, which is a big difference with big life-history implications. Is such intra-specific variation expected? Or does it reflect difficulties and huge uncertainties in fitting the VB curve to potentially low quality age-length datasets? Either way, this should be explained and justified. Given that the projected differences in Linf and K due to 1C warming are in the order of 2-5% (and are apparently having a big effect on population growth rate, but see my comment below), 2-3 fold intra-specific differences across studies may be of concern.

I like the authors' approach of mixed effects models. The effects on Linf and K are interesting and fit with the general expectations. I am less convinced about the natural mortality, M, because it is not an independent measure but is estimated from K and Linf. Looking at Fig2 a) and c) and model fitting in Table S1 one can see that trends and variation for K and M are very similar. For other life history variables the original data is very limited and in many cases has to be estimated from Linf and K using many assumptions. So in the end the number of life-history variables explored in this study is not that big; basically it is Linf and K, and even they are not independent from each other because they are estimated from fitting length-age data to one model.

While the mixed effect models are a good and transparent way to summarise the assembled data, the projection of species responses using life-history tables is more debatable. The life-history tables themselves are of course a widely accepted tool to estimate population growth rates, but the estimates entirely depend on age-specific maturity, and estimates of mortality and fecundity. None of these estimates are really available for the studied species. Therefore:

1) mortality estimates used here entirely depend on K and Linf and the same equation is applied to all species ($M = \exp(0.55 - 1.61 \cdot \log(L) + 1.44 \cdot \log(\text{Linf}) + \log(K))$). This equation for mortality might be ok for broad scale inter-species comparisons but it is highly questionable whether it

would apply in case of small intra-specific changes. Do the authors have evidence to support the application of this equation to intra-specific analyses?

2) Age at maturity estimates are also dependent on K and L_{inf} (equation 2) using general inter-species derived equations (same reservations apply)

3) Fecundity relates to length identically in all species (equations 4 and 5)

This means that the estimates of population growth rates are entirely dependent on the estimates of L_{inf} and K , and the outcomes of the model are relatively easy to predict. Smaller and shorter lived species have reduced fecundity (due to decreasing L_{inf}) but don't have enough time to benefit from reduced mortality (applying M equation above shows that 5% reduction in L_{inf} and 5% increase in K leads to 3% decrease in mortality across all lengths), compared to long-lived species that have same reductions in fecundity and have more time to benefit from reduced mortality. In reality it is likely to expect that increased mortality will lead to increased reproductive investment, which might increase population growth rates.

Finally, the life-history table approach also assumes that fecundity has linear relationship to recruitment, i.e. there are no differences in egg sizes among species and that a certain change in fecundity will directly translate into the same change in recruitment. Such application of life-history tables might be justifiable for say, birds or mammals, or other species where fecundity has close correlation to the realised offspring numbers. However, in fish this is highly questionable, and at least this assumption should be clearly explained in the main text. Many fisheries biologists would argue whether a 2% decrease in fecundity leads to a 2% decrease in recruitment.

In summary it might be worthy to present such general life-history expectations of population growth responses, as it may encourage future empirical analyses and testing. However, the limitations and assumptions of the approach should be more explicit in the main text itself.

Reviewers' comments:

Reviewer #1 (Remarks to the Author):

This study quantifies the relationship between ocean temperature and seven life history traits for 332 fish species occurring in the Pacific and Indian Ocean basins and the expected cumulative impacts of these relationships on population growth rates under a degree of warming. The authors individually regress each life history trait against temperature. They then use life tables to derive and compare the intrinsic rate of population growth at average temperature versus one degree warmer than average by using their individual regressions to predict life history traits at each of those temperatures.

My primary concern is that the authors predict the impact of temperature on each life history trait individually (seven univariate models) rather than in a single multivariate model. In fact, life history traits are correlated to each other in addition to temperature and the univariate approach does not account for this covariance. Thorson et al. (2017) use a multivariate phylogenetic model to predict the relationships between temperature and a highly similar set of life history traits for all fish species using life history data from FishBase. The authors should use this or a similar modeling approach to re-estimate the influence of temperature on life history accounting for correlations between traits and across phylogenetic relatedness. Fortunately, Thorson et al. (2017) provide an R package that allows their model to be refit to a new set of data: <https://github.com/James-Thorson/FishLife>

Thorson, J. T., S. B. Munch, J. M. Cope, and J. Gao. 2017. Predicting life history parameters for all fishes worldwide. *Ecological Applications* 27(8): 2262-2276.

Our analysis accounts for the phylogenetic relatedness because we used linear mixed-effect model to account for species- and family-related random variance. We have rewritten L87-89 to provide more clear explanations about our analysis.

We had already used a multivariate method (Principal Component Analysis), but we have expanded this part and made it more visible. To account for the life history correlation, we conducted 3 PCA ordinations, each with a set of traits: (1) $\ln K$, $\ln L_{\infty}$, $\ln M$, and $\ln b$, (2) $\ln L_{50}$ and $\ln A_{50}$, and (3) $\ln K$ and $\ln L_{\infty}$. Because PCA is ideally conducted on correlation matrices without missing data, including more traits leads to fewer observations; a combination of all 7 traits results in so small sample size

(n=15) that we dropped the trait $\ln A_{\max}$ altogether. We then conducted linear mixed-effect models to evaluate the effect of temperature on the first principal components (PC1) of these ordination analyses (see Table S2, L103-105). The result is consistent with the univariate main results: increasing temperature is associated with PC1 score moving in “faster” direction.

Furthermore, as suggested by reviewer 1, we also investigated the mean life history responses to temperature using the multivariate model by Thorson et al. (2017). Thirty of our 332 species are not present in the Fishbase. Excluding these species, we estimated the covariance coefficient between temperature and each of the six traits: L_{∞} , K , t_{\max} , t_m , M , and L_m . The signs of mean covariance coefficients of these traits are generally consistent with those of the fixed-effect slopes in our empirical analysis (compare Table S1 vs. Table S3). However, we cannot compare magnitudes of these mean covariance coefficients with the slopes as temperature scales are different between the model by Thorson et al. (2017) and our empirical analysis. We have added a table of mean and 95% CI of covariance coefficients for the six traits in the Table S3 and description of consistency between model and empirical analysis in L105-107.

In summary, including the multivariate perspective is not changing our main conclusions, but is making them stronger.

My secondary concern is the extent to which these results can be used to predict cumulative impacts on the intrinsic growth rate on the entire population given the following argument: if populations are shifting distributions to follow their preferred temperatures shouldn't the average temperature remain the same through time and thus no result in a change in population-wide intrinsic growth rate (just changes in latitude-specific growth rates)? To make this argument hold water, the authors will need to present the evidence that as populations shift distributions in response to climate change, the mean temperature will increase or the distribution of temperatures will increase such that the spatial distribution shifts don't equalize impacts on life history.

We agree shifting distribution is one way of coping with temperature change. However, some populations are more stationary than others, and we generally do not expect populations to exactly track changing temperature isoclines. We have revised the text to make it clearer that as warming persists, it can affect life history traits that lead to demographic or abundance changes in fishes along with their shifts

in distributions (Punzón et al. 2016; Poloczanska et al. 2016). Please see L 46-48 and L213-222.

Related to this concern is the fact that the influence of temperature on life history traits is dome-shaped and not monotonic and that neither the univariate or multivariate approaches account for this. Ultimately, I think it's okay because shifting distributions will prevent mean temperatures from moving too far past the thermal optimum but I think this should be acknowledge in the manuscript. For example, in Figure 2, growth rate can't continue to increase forever under increasing temperatures, it will decrease past some threshold.

We evaluated the nonlinear temperature effects on life history traits using the generalized additive mixed-effect models (GAMM). Consistent with our present results, the GAMM results indicate approximately linear temperature effects on life histories within ± 5 °C from the mean temperature. We added these results into Figure S3 and Table S4.

With the combined linear and nonlinear model results, we added a sentence to describe the potential nonlinear responses of life history traits when temperature exceeds species thermal optima in L 109-114.

My final big concern is the straight vertical line of data points in Figure 2 – what is going on there?

The x-axis represents centered temperatures by species. Thus, data points of the populations with habitat temperature close to mean temperature are located in the middle of x-axis. The vertical line of points at mean temperature (0 °C) is because some species are represented by single populations, and because our database comprises a large number of reef fish populations with narrow ranges of habitat temperatures.

Small comments

25 – what's the connection between reproductive rates and asymptotic rate and age-at-maturation? Should these be separate traits in the list?

We have separated the asymptotic length and age at maturation into 2 parentheses as suggested (L24-25).

27 – delete colon and make separate sentence

Done.

27 – species groups appears to be based on habitat associations -- reword to convey this

We have reworded “species groups” into “habitat-related species groups”. L26.

28 – move this sentence summarizing results after the sentence summarizing methods (the life table sentence)

Done.

31 – age-specific

We have removed this term.

34 – and lower proportions in what groups?

We have removed this line.

40 – ocean warming isn’t contributing to acidification – reword

Done. “Climate change is projected to increase ocean temperature along with driving other physical and biogeochemical changes such as acidification and expansion of hypoxic zones.”. L39-41.

50 – there’s been lots of work on temperature effects of recruitment and on somatic growth rates that is not cited here

Done. We added Sponaugle et al. 2006; Neuheimer et al. 2011; Kuparinen et al. 2011. L52-53.

51 – Rewrite to say that temperature can produce intraspecific variation in life history

Done. L53.

61 – positive allometric

Done. L63.

64 – The study question, region, and methods have not been introduced at all. You need a paragraph introducing the purpose of the study and general methods.

Done. L69-74.

69 – this is half the world’s oceans – saying that it’s where some of the most rapid warming has occurred isn’t that interesting or profound. And isn’t it not true? Isn’t the Northeast US and the southern Ocean the fastest warming?

Agree. We have corrected this sentence into: “This region includes several major ocean warming hotspots (24).”. L73.

79 – derived temperature variables from what dataset? At what time of year? Is it matched to the time the life history trait was measured?

Done. L81-87. Because temporal changes in temperature were small compared to the geographic gradient in temperatures among populations, we used long-term average data from NOAA’s WOA 13 to derive temperature indices for our analysis.

88 – I think you should state that this is a life-history invariant method so that the reader knows it was derived from other life history variables

Done. L95.

92 – across all species

Done. L99.

98 – would be useful to see sample sizes somewhere

We have cited Table S1 to provide sample sizes for these models (L116).

104 – why did you define species groups based on habitat affiliation and not some other way of grouping species (trophic level, taxonomic order, functional group, etc). As contrasting environments may induce variation in life history responses (Jonsson and Jonsson 2011), we evaluated life history responses among habitat-related functional groups. L120-122.

104 – define groups here

Done. Table S5, L122.

132 – Rewrite the sentence to say that use a life table to “insert what you use life table to do”. This will more clearly set up readers expectations for what the purpose the life table is.

We have added short description about warming-induced effects on demographic rates and the expected population growth based on reviewer 3’s suggestion (L151-157). Then, we stated the purpose of using life table: “To understand the temperature-related demographic consequences on population growth, we use a life table model...” as suggested by reviewer 1 (L158-162).

144-146 fix grammar of this sentence

We have removed this sentence.

150 – where do you show that populations in warmer habitats were more negatively impacted? I don’t see this evidence presented.

We revised this sentence. L180-183.

185 – I think this grossly ignores the Thorson et al. paper that examines the influence of temperature on all of these life history traits simultaneously across all taxa. In general, this work needs to be better acknowledged throughout the paper.

We now cited Thorson et al. 2017 as suggested. L229.

201 - this study doesn’t say there has be no overall effect. It says there have been some winners, some losers, and some unaffected, and that the losers outweigh the winners.

We fixed this line into: “Given that recent research found a mix of positive, negative, and neutral effects of climate warming on population sustainability, ..”. L244-245.

342 – how would your life history trait database look differently if you had drawn the life history data from FishBase using the rfishbase R package? Do you have more data? Less data?

FishBase contains records of 33,104 species, among which 32,968 species have only 1 population record. Excluding these single-record species, multiple population

records of life histories are available for only 136 species. On the other hand, our data include 332 species, among which 206 species have multiple population life history data. Consequently, our data have more population records than Fishbase. With the fewer population records of life history data in Fishbase, it may lead to a bias or reduced statistical power for detecting temperature effects on within-species life history response.

We added 2 sentences to describe difference in the distribution of population records per species between the Fishbase and our data. L402-403, 415-416.

362-365 – If these natural mortality estimates aren't comparable, why do you include them together in the same regression?

We re-estimated M for individual populations using a life-history invariant method (model II of Gislason et al. 2010). See L440-442. We used the Gislason M estimates to conduct the regression and life table modeling analysis. L495-518.

393-387 – What is the temporal resolution matched to the life history data? The mean decadal temperature for that cell? Can you clarify this?

The mean decadal temperature was matched to each of the life history datum in a cell as suggested by reviewer 1. We clarified this in L478-479.

435 – I'm curious about these species. Are the squids? Shrimps?

We excluded estimates of M for 3 species of round herring: *Spratelloides delicatulus*, *Spratelloides gracilis*, and *Spratelloides lewisi*.

436 – Plot histograms of life history parameters in supplement?

We added histograms of all life history parameters in Fig. S1 as suggested.

500-502 – I thought the whole point is that a rise in temperature has different effects on different species? Is this the mean?

Previously, we explored temperature-induced effects on population r-ratios based on mean life history changes (i.e., the slopes of fixed-effect variables). Upon reviewer 1's question, we found that we could have overlooked species differences in responses to warming with these mean temperature effects. Consequently, we tried to incorporate differential species slopes of such temperature effects in the revision. Surprisingly, we found that the projected population r-ratios changed greatly when considering species-specific and family-specific temperature effects (revised Fig 4 and supplemental Fig. S5). These changes reflect a novel aspect undetected in our previous analysis; i.e., in addition to natural mortality and fertility, rising temperature also leads to differential reduction in generation time, with greater reductions for species with fast life histories compared to those with slow life histories. Taking into account temperature effects on natural mortality, fertility, and generation time, we

found that populations r-ratios show a declining relationship with generation time (Fig 4).

We thank reviewer 1 for suggesting to account for species-specific responses in life history traits under warming, as this helps us discover the temperature effect on generation time.

Table and Figure comments

All of the figures require lots of cosmetic work to make them up to the standards of a Nature journal.

Fig 2 – what are the straight lines at zero? Color by group and show group fixed effect? What about random effect by group since you have that hypothesis? No p-values shown here. Mark significance. Why no lines for some? Why monotonic and not dome shaped

Models for Fig. 2 do not include groups as a fixed- or random-effect predictor. Consequently, we do not color lines by groups. We show the effects of groups in Fig. 3 and Table S6.

Please see our response to major comment 4. The x-axis represents centered temperatures by species. Thus, data points of the populations with habitat temperature close to mean temperature are located at 0 of x-axis. The vertical line of points at mean temperature (0 °C) may be due to that presence of some single-population species and a large number of reef fish populations with narrow ranges of habitat temperatures in our database.

We now add tendency lines for all panels and use different line types (solid vs. dashed lines) to show significance of the fixed effects of centered SST on each life history trait per reviewer 1's suggestion. See Fig. 2. We also add analysis of nonlinear temperature effects on life history traits in Fig. S3. See our response to major comment 3.

Fig 3 – Why is intercept interesting? Why not remove the intercept plots and show the slope plots for all of the evaluated life history parameters instead?

We have modified the scales for the intercepts, demonstrating interpreting these intercepts in the units of L_{∞} , K , and M , respectively. As these intercepts show pronounced differences in the temperature-independent trait values among groups

(see Fig. 3), we decide to include the data of intercepts in the PCA analysis.

As our database does not have sufficient population data for the b , L_{50} , A_{50} , or *lifespan*, we could not include slopes of the temperature effects on these traits in the PCA (see Table S6 d-g).

Fig 4 – In caption, define what values above and below zero mean. Explain the bimodal shape in panels A and B. Color points by habitat association?

We revised Fig 4. Panels a and b now show linear relationships between population r-ratios and generation time. We have colored points by habitat association for panels a and b as suggested.

Figure S1 – how is a population defined and delimited?

We now added definition of a population: stock in different sites are considered as different populations of a given species. L416-417.

Reviewer #2 (Remarks to the Author):

This study appears to show that life-history responses to warming differ in relation to phylogeny and habitat with reef and demersal fishes being more sensitive to temperature change than bathydemersal fishes. Interestingly, rising temperature while promoting faster life-history traits, seems to have positive or negative influences on population growth, depending on populations' position on a life-history continuum. The evidence presented that fast life-history populations are more vulnerable to warming compared to slow life-history ones is particularly important in the debate over climate change impacts.

Thank you.

The authors attempt to simultaneously account for multiple traits in predicting the consequences of warming which is an important advance over many published studies that have only considered single traits. The number of species and populations considered is also impressive. The methodological approach taken seems reasonable and appropriate given the type of data available for so many species. The statistical analysis using LME for comparing LH traits and temperature and the PCA seem appropriate and the results are clearly explained.

Considering natural mortality for so many species is a problem to any large comparative study so I understand why the authors have used the Gislason et al. (2010) relationship. While the authors of that study did suggest that temperature

could be used to explore some aspects of warming, I am not clear as to why estimation of M for the length at age-at-50% maturity provides a good basis for inter-species comparison.

Population M is a function of L_{∞} , K , and body length of based on Gislason et al. (2010). In this study, body length is measured as the midpoint of length range of a population. However, as the length range data are not available to us, we could not determine the midpoint of lengths for our study populations. On the other hand, a study by Charnov et al. (2013) suggests a invariant relationship exists among mortality at age of maturity, length at maturity, L_{∞} and K ; i.e., $M_{\alpha} = \left(\frac{L_{\alpha}}{L_{\infty}}\right)^{-1.5} * K$.

Referring to this, we used the length at age of maturity as an alternative estimate of body length in the Gislason model II to estimate population natural mortality. We have provided this explanation in the Methods (L 503-506).

Although I appreciate why SST and BT were used as proxies I did not understand why only model fits for the relationships between mean SST and each life history trait by six groups of fishes were presented in Table S2, as while SST may be relevant for pelagic fish it won't be for bathydemersal species.

We have added the model fit with predictor BT into Table S6 (h-n) based on reviewer 2's request.

While I think the major claims of the study will need to be explored in greater detail at the species or guild level, I think this study does provide an important comparative advance that will interest climate scientists and fishery managers.

Thank you.

Minor comments

Did the authors really mean to say that fish adapt to temperature increases by displaying faster life- history strategies such higher mortality?

Many studies demonstrate temperature-dependent life history variation among populations within a species. In general, populations display faster growth, earlier maturation and higher mortality (or shorter lifespan) at warmer temperature compared to cooler temperature. We agree with reviewer 1 that such temperature-dependent trait variation may be mediated by recruitment, changes in somatic growth rates, or adaptive mechanisms. We have cited some references to about multiple explanations for temperature-life history correlations (L53-55).

Given that temperature is not the only driver of long term life history traits, compiling life history data from 1958 – 2017 may partially compromise the

comparative analysis. I would like to hear the authors view as to whether this is important to the study. Personally, I doubt it will be a major weakness of the study as differences among taxa are generally greater than intra-specific variation. As authors say on line 102 there significant species- and family-related variance was found for most life-history traits.

Thank you for your suggestion. We agree that temporal changes in such as fishing intensity or additional environmental effects could potentially confound the long-term temperature effects on within-species life history traits. Potentially, we could include year as an additional variable into our analysis to evaluate significance of temporal effects. However, because it requires a large amount of work to compile data on year, and as our meta-analysis has accounted for species-and family-related variance, our results should be robust without the year effect.

Reviewer #3 (Remarks to the Author):

This study explores the relationship between water temperature and several life-history characteristics in a large literature derived dataset of over 300 fish species from Indian and Pacific Oceans. To assess how populations of these species are likely to respond to warming the authors apply life-history tables to estimate population growth rates under changing life-history characteristics.

I applaud the authors for compiling this dataset and would strongly urge that upon the publication the dataset is made public with FULL references included (at the moment the dataset I see only has year and author name). However, my understanding is that the authors use Von Bertalanffy growth curve coefficients K and L_{inf} from comparable models (line 361) without assessing the quality of the actual dataset to which the curves were fit. I realise such specific checking would perhaps be too big of a task, however, fitting VB curves is not straightforward, and the procedure is sensitive to the absence of oldest (in case of heavily fished stocks) or youngest (in case of fishery dependent datasets) age classes in the data. My quick look through the data set shows large amounts of intra-specific variation. For example, entry 11 shows K and L_{inf} of *Trichiurus lepturus* to be at 0.41 and 61.5cm, whereas entry 6 for the same species (and not that distant a geographic location) shows 0.116 and 193.8 cm. Is such a 3-fold difference in L_{inf} reasonable? If I filter species that have at least five entries and calculate the ratio between min and max L_{inf} values for each species, I find that 20 out of 81 species have ratios >2 (one species has a ratio of 11). For the coefficient K , half of the ratios are >3 . This would correspond to values of, say, 0.05 and 0.15 for the same species, which is a big

difference with big life-history implications. Is such intra-specific variation expected? Or does it reflect difficulties and huge uncertainties in fitting the VB curve to potentially low quality age-length datasets? Either way, this should be explained and justified. Given that the projected differences in L_{inf} and K due to 1C warming are in the order of 2-5% (and are apparently having a big effect on population growth rate, but see my comment below), 2-3 fold intra-specific differences across studies may be of concern.

We are aware of the impact of data on the fit of von Bertalanffy model, and thus, we have excluded some data of relatively poor quality (see L 453-462). However, after such data scrutinizing, the data still contain high variability in K and L_{∞} within some species, as suggested by reviewer 3.

Presence of large intra-specific variability in the K and L_{∞} may be partially due to that the estimation of these traits depends on one another. To merge these two correlated traits into a single trait, we conducted PCA for $\ln K$, and $\ln L_{\infty}$ (see our response to the major comment 1 by reviewer 1) and evaluated the ratio of maximum-to-minimum scores of the PC1 among species. We found the absolute values of ratios of $PC1 < 3$ for most species ($n=285$), with the $|ratios| \geq 3$ for 34 species (see Figure S7). Thus, we compared the LMEs of temperature effects on individual life history traits with data of all populations vs. those after excluding the 34 species with large ratios of PC1 (Table S9). This result is consistent with our present result in Figure 2 and Table S1. We conclude that even though there can be issues with some estimates, our results are robust and not driven by such anomalies. We have provided the explanations above in L544-554.

I like the authors' approach of mixed effects models. The effects on L_{inf} and K are interesting and fit with the general expectations. I am less convinced about the natural mortality, M , because it is not an independent measure but is estimated from K and L_{inf} . Looking at Fig2 a) and c) and model fitting in Table S1 one can see that trends and variation for K and M are very similar. For other life history variables the original data is very limited and in many cases has to be estimated from L_{inf} and K using many assumptions. So in the end the number of life-history variables explored in this study is not that big; basically it is L_{inf} and K , and even they are not independent from each other because they are estimated from fitting length-age data to one model.

Thanks. Although our estimates of M depend on the K and L_{∞} , our finding of the temperature-dependence of M was consistent to those in previous studies using empirical estimates (Pauly 1980; Gislason et al. 2010) or predicted M based a

multivariate model (Thorson et al. 2017; Table S2). We have described consistency in temperature-dependence of M between the empirical and model analyses based on reviewer 1's major comment 1. See Table S3, L105-107.

While the mixed effect models are a good and transparent way to summarise the assembled data, the projection of species responses using life-history tables is more debatable. The life-history tables themselves are of course a widely accepted tool to estimate population growth rates, but the estimates entirely depend on age-specific maturity, and estimates of mortality and fecundity. None of these estimates are really available for the studied species. Therefore:

1) mortality estimates used here entirely depend on K and L_{inf} and the same equation is applied to all species ($M = \exp(0.55 - 1.61 \cdot \log(L) + 1.44 \cdot \log(L_{inf}) + \log(K))$). This equation for mortality might be ok for broad scale inter-species comparisons but it is highly questionable whether it would apply in case of small intra-specific changes. Do the authors have evidence to support the application of this equation to intra-specific analyses?

Hutchings and Kuparinen (2017) applied a similar estimation method for M (Charnov et al. 2013), also based on the life history invariant concept, for their intra-specific analysis. Because our stock-specific estimates of M based on Gislason et al. (2010) model are correlated with those based on Charnov et al. (2013), our estimates of M could also be used for intra-specific analysis.

We do not claim that our life tables are necessarily very accurate, in particular because of paucity of data on recruitment. This is the reason why we look at differential effects through r -ratios instead, such that the early life history effects are partially cancelled out. There is also uncertainty regarding the later life history. Nevertheless, we argue that the life table analysis is still a step forward in interpreting implications of single trait changes. Indeed, we are not aware of any alternative methods to provide such insights under data-poor situations (see L213-222).

2) Age at maturity estimates are also dependent on K and L_{inf} (equation 2) using general inter-species derived equations (same reservations apply)

We used 1) available empirical age-at-50% maturity (A_{50}) data for 100 populations and 2) K and L_{∞} -derived A_{50} data to conduct life table projection (estimating population r -ratios, Fig. 4a, b). See L593-597. Results of the life table projection are consistent between the use of different A_{50} data, indicating that our result does not depend on the assumption of direct association between K and L_{∞} and A_{50} .

3) Fecundity relates to length identically in all species (equations 4 and 5)

This means that the estimates of population growth rates are entirely dependent on the estimates of L_{inf} and K , and the outcomes of the model are relatively easy to predict. Smaller and shorter lived species have reduced fecundity (due to decreasing L_{inf}) but don't have enough time to benefit from reduced mortality (applying M equation above shows that 5% reduction in L_{inf} and 5% increase in K leads to 3% decrease in mortality across all lengths), compared to long-lived species that have same reductions in fecundity and have more time to benefit from reduced mortality. In reality it is likely to expect that increased mortality will lead to increased reproductive investment, which might increase population growth rates.

We have revised life table projection analysis to account for temperature-induced species-specific responses based on reviewer 1's suggestions (see revised Fig. 4 and Fig. S5). Our updated model projection shows that warming effects on population growth depend on generation time, as shown by a negative relationship between population r-ratios and generation time. This indicates greater impacts of warming on species with slow life histories. L169-183.

Finally, the life-history table approach also assumes that fecundity has linear relationship to recruitment, i.e. there are no differences in egg sizes among species and that a certain change in fecundity will directly translate into the same change in recruitment. Such application of life-history tables might be justifiable for say, birds or mammals, or other species where fecundity has close correlation to the realised offspring numbers. However, in fish this is highly questionable, and at least this assumption should be clearly explained in the main text. Many fisheries biologists would argue whether a 2% decrease in fecundity leads to a 2% decrease in recruitment.

We now provided explanations about that our use of life table to derive lifetime spawning biomass, an important determinant for the recruitment strength (L158-162). Due to unavailable stock-specific recruitment data, we predicted temperature effects on population growth with the assumption of no changes in recruitment (L162-165).

In summary it might be worthy to present such general life-history expectations of population growth responses, as it may encourage future empirical analyses and testing. However, the limitations and assumptions of the approach should be more explicit in the main text itself.

We now described the expected responses of life history traits and population growth under warming, and we clarify limitations, assumptions, and interpretation with care for the life model projection as suggested by reviewer 3. See L151-165, 213-222.

Peer Review File –

Reviewers' comments second round

Reviewer #1 (Remarks to the Author):

Major comments

I appreciate the author's efforts to address my concerns from the first review and found the new text additions and figures helpful. For example, from my big concerns before, there explanation of the pile up of points at zero was clear and clarified my confusion and the GAMMs exploring potential non-linear results were very helpful and interesting.

I was less satisfied in how they addressed my other two major concerns which concerned the exclusion of FishBase life history approach and the consideration of using a more sophisticated multivariate approach such as that used by Thorson et al. (2017) and recently again by Thorson (2020). I am prepared to drop the multivariate model issue but still want a more clear explanation about why FishBase life history is not considered,

I am still curious about why the authors have ignored the vast amount of life history data available in FishBase. Their assertion that only 136 species in FishBase have life history estimates for multiple populations is demonstrably incorrect. If you run the following code, you will be able to access all of the FishLife life history for growth and maturity and will see 100s to 1000s of species with more than 2 records.

```
# Packages (both can be installed from CRAN)
library(tidyverse)
library(rfishbase)

# Get FishBase fish species
fb_fish <- rfishbase::load_taxa(server = "https://fishbase.ropensci.org")

# Get FishBase life history information
fb_fish_vonb <- rfishbase::popgrowth(species_list=fb_fish$Species)
fb_fish_maturity <- rfishbase::maturity(species_list=fb_fish$Species)

# Calculate number of Von B records by species
fb_fish_vonb_stats <- fb_fish_vonb %>%
  group_by(Species) %>%
  summarize(n_k=sum(!is.na(K)),
            n_linf=sum(!is.na(TLinfinity)),
            n_m=sum(!is.na(M)))
sum(fb_fish_vonb_stats$n_k>=2)
sum(fb_fish_vonb_stats$n_linf>=2)
sum(fb_fish_vonb_stats$n_m>=2)

# Calculate number of maturity records by species
fb_fish_maturity_stats <- fb_fish_maturity %>%
  group_by(Species) %>%
  summarize(n_tmat=sum(!is.na(tm)),
            n_lmat=sum(!is.na(Lm)))
sum(fb_fish_maturity_stats$n_tmat>=2)
sum(fb_fish_maturity_stats$n_lmat>=2)
```

Why ignore more data when it is so easily available? Each of these traits is tied to a reference where coordinates might be possible and the traits have meta-data about how they were calculated. Is my confusion related to your definition of a "population"? Which incidentally also needs to be more clearly defined in the manuscript. Anyways, FishBase seems like a rich and easy

resource and like it should be a source to supplement the Google Scholar and DFO results.

The authors should be aware that Jim Thorson's FishLife now predicts intrinsic population growth rate for all fish species accounting for phylogenetic relatedness and covariance among traits including temperature and like it could be used to predict changing intrinsic population growth rate with changing temperature.

Thorson, J.T., 2020. Predicting recruitment density dependence and intrinsic growth rate for all fishes worldwide using a data-integrated life-history model. *Fish and Fisheries*, 21(2), pp.237-251.

I don't understand this well enough to mandate that the authors take this approach but I think they should acknowledge this paper.

Minor comments

24 - (but with decreasing asymptotic length)

25 - (but with decreasing age-at-maturation)

29 - mortalities, fecundities, and generation times

30 - confused by results presented from 30 to 37

31 - change "will gain an advantage of" to "will experience faster population growth rates through"

33 - declines in what? growth rate?

53 - "Warmer environments are typically associated with smaller body size, higher mortality, faster growth, and earlier maturation (13-16)."

58 - This sentence feels out of place. I suggest deleting: "or anthropogenic effects. Consequently, life-history traits inform the reference points for fisheries species and likelihood of recovery for the overfished populations (20, 21)."

75 - Delete this sentence to improve flow between paragraphs.

77 - what defines a population?

79 - "habitat temperature data are available from NOAA' World Atlas 2013"

85 - "Because the recent rate of change in the ocean temperatures in our study region has been minor (< 2 C per 100 years, 24, 26), we did not account for the small temporal changes in the past sea temperatures." I don't understand this sentence and suggest it be deleted.

100 - make all plural: "higher natural mortality rates (M), smaller asymptotic lengths (Ling), etc"

123 - K, Linf, and M values (added Oxford comma)

159 - you called it a "life-table model" in the abstract. be consistent throughout.

175 - K, Ling, and model-derived A50 (added Oxford comma)

182 - reword as "but negative impacts on fish populations with slow life histories, especially elasmobranchs."

196 - "refuting in potential declines in population growth rate" - be very careful not to conflate population growth rate with population size - re-examine the MS to make sure this doesn't happen else where.

218 - suggest removing this text which feels tagged on "plasticity in spatial distribution is a mitigating factor to such temperature sensitivity." However, thanks for adding the previous text in response to my earlier comment.

222 - "all but a few"

228 - "primarily focused"

231 - "population growth rates"

244 - The summary of this study remains incorrect. It measured the impact of historical warming on population growth rates (r , intrinsic growth rate) and found that some populations have been positively and negatively influenced by warming. This is similar to your finding that some life histories should benefit from warming while others should be negatively impacted. Different from your study which is entirely based on life history, its estimates are also subject to ecosystem effects (match-mismatches) and fisheries induced changes in age structure. The point is that "population sustainability" is incorrect - it measured impacts on "population growth rate" -- and it didn't contradict other models (the historical patterns differ from forecast ones).

Figure 2 - I understand now why there are so many points at zero from your clear response. Can you add a sentence explaining this in the caption?

401 - This is incorrect. See comment and code above.

Reviewer #2 (Remarks to the Author):

The authors have dealt with most of my comments and the study should encourage future analyses of population growth responses. In any wide ranging comparative study like this there is always going to be a limitation on the accuracy of data but the authors have made sensible attempts to consider the robustness of their analyses.

The addition of the model fit with predictor BT into Table S6 (h-n) as I requested does appear to weaken the argument on Line 127 "of consistent signs of temperature effects on the mean responses of K, M, and for all 6 groups groups—positive slopes for K and M but negative slopes for Linf (Fig Fig. 3a–c; Supplemental Table S6)—even though some of these effects were not significantly different from zero, particularly for the pelagic and bathydemersal fishes (Fig. 3a–c)". I therefore think some mention is made of this with respect to the relevant temperature measures for demersal and bathydemersal fish.

Line 20 . The authors may have misunderstood my point. I still don't think this sentence makes sense: Most marine fish species adapt to temperature increases by displaying higher mortality. Fish may be become subject to greater mortality if predation and disease increases. The sentence on Line 53 does make sense as warmer environments are often associated with high mortality.

Reviewers' comments:

Reviewer #1 (Remarks to the Author):

Major comments

I appreciate the author's efforts to address my concerns from the first review and found the new text additions and figures helpful. For example, from my big concerns before, there explanation of the pile up of points at zero was clear and clarified my confusion and the GAMMs exploring potential non-linear results were very helpful and interesting.

I was less satisfied in how they addressed my other two major concerns which concerned the exclusion of FishBase life history approach and the consideration of using a more sophisticated multivariate approach such as that used by Thorson et al. (2017) and recently again by Thorson (2020). I am prepared to drop the multivariate model issue but still want a more clear explanation about why FishBase life history is not considered,

I am still curious about why the authors have ignored the vast amount of life history data available in FishBase. Their assertion that only 136 species in FishBase have life history estimates for multiple populations is demonstrably incorrect. If you run the following code, you will be able to access all of the FishLife life history for growth and maturity and will see 100s to 1000s of species with more than 2 records.

```
# Packages (both can be installed from CRAN)
library(tidyverse)
library(rfishbase)

# Get FishBase fish species
fb_fish <- rfishbase::load_taxa(server = "https://fishbase.ropensci.org")

# Get FishBase life history information
fb_fish_vonb <- rfishbase::popgrowth(species_list=fb_fish$Species)
fb_fish_maturity <- rfishbase::maturity(species_list=fb_fish$Species)

# Calculate number of Von B records by species
```

```

fb_fish_vonb_stats <- fb_fish_vonb %>%
group_by(Species) %>%
summarize(n_k=sum(!is.na(K)),
n_linf=sum(!is.na(TLinfinity)),
n_m=sum(!is.na(M)))
sum(fb_fish_vonb_stats$n_k>=2)
sum(fb_fish_vonb_stats$n_linf>=2)
sum(fb_fish_vonb_stats$n_m>=2)

# Calculate number of maturity records by species
fb_fish_maturity_stats <- fb_fish_maturity %>%
group_by(Species) %>%
summarize(n_tmat=sum(!is.na(tm)),
n_lmat=sum(!is.na(Lm)))
sum(fb_fish_maturity_stats$n_tmat>=2)
sum(fb_fish_maturity_stats$n_lmat>=2)

```

Why ignore more data when it is so easily available? Each of these traits is tied to a reference where coordinates might be possible and the traits have meta-data about how they were calculated. Is my confusion related to your definition of a “population”? Which incidentally also needs to be more clearly defined in the manuscript. Anyways, FishBase seems like a rich and easy resource and like it should be a source to supplement the Google Scholar and DFO results.

Thanks for helping clarify the rfishbase data. When we checked rfishbase in the previous revision in Dec 2019, there were 33,104 species in total. Using the codes above, we found that number of species in rfishbase has increased to 43,302. We also used the codes below to check available population records with temperature for a subset of life history traits in rfishbase:

```

> fb_fish_vonb1 =
rfishbase::popgrowth(species_list=fb_fish$Species,fields=c("SpecCode","Species","StockCode","Loo","K","Temperature"))
> fb_fish_vonb2 =fb_fish_vonb1[is.na(fb_fish_vonb1$Temperature)==FALSE,]
> fb_fish_vonb2$y=1
> dim(xtabs(y~SpecCode,fb_fish_vonb2)[xtabs(y~SpecCode,fb_fish_vonb2)>=2])
[1] 916
> kept2 = unique(fb_fish_vonb2$SpecCode)[xtabs(y~SpecCode,fb_fish_vonb2)>=2]
> fb_fish_vonb3 = fb_fish_vonb2[fb_fish_vonb2$SpecCode %in% kept2,]

```

```
> dim(fb_fish_vonb3)
```

```
[1] 5410 6
```

The table below summarizes available population records with temperature in rfishbase vs. our data. We found more data in rfishbase for K , L_{∞} , L_{50} , and A_{\max} , but not A_{50} (Note that availability for M is not comparable between 2 datasets due to differences in estimation methods). These results show that rfishbase provides a rich resource to supplement to our data as suggested.

Trait	no. sp in rfishbase (w/ temperature)	no. pop records in rfishbase (w/ temperature)	no. pop records in our study (w/ SST) (Table S1)
K and L_{∞}	916	5410	1268
M	170	483	1268
A_{50}	12	30	118
L_{50}	196	576	162
A_{\max}	112	332	194

Furthermore, we found that one can access locality and references but not spatial coordinates in rfishbase. This indicates difficulty to recover missing temperature data for the population records of A_{50} . Consequently, we now add 2 sentences to acknowledge rfishbase and provide reasons for constructing our data in the Methods: “Although similar data are available in *FishBase*, a global and freely-available database on fish, we opted not to use data from *FishBase* because temperature data associated with populations are often missing and difficult to restore due to lacking spatial coordinates in *FishBase*.” L421-425.

The authors should be aware that Jim Thorson’s FishLife now predicts intrinsic population growth rate for all fish species accounting for phylogenetic relatedness and covariance among traits including temperature and like it could be used to predict changing intrinsic population growth rate with changing temperature.

Thorson, J.T., 2020. Predicting recruitment density dependence and intrinsic growth rate for all fishes worldwide using a data-integrated life-history model. *Fish and Fisheries*, 21(2), pp.237-251.

I don’t understand this well enough to mandate that the authors take this approach but I think they should acknowledge this paper.

Thanks for providing this reference. We now cite Thorson 2020 as suggested. L61 and 166.

Minor comments

24 - (but with decreasing asymptotic length)

Done.

25 - (but with decreasing age-at-maturation)

Done.

29 - mortalities, fecundities, and generation times

Done.

30 - confused by results presented from 30 to 37

We have reworded these sentences. "Using a life-table model, we show that the combined effects of temperature-induced faster early-life growth but lower asymptotic size, together with reduced mortality and increased fecundity during younger ages, tend to facilitate population growth for slow life-history populations, but reduce it for fast life-history ones. Furthermore, our model predicts declines in population growth rates for lower proportions (25–30 %) of slow-life history fishes but that for greater proportions of fast-life history fishes (42–60 %) within our data under 1° C warming.". L28-35.

31 - change "will gain an advantage of" to "will experience faster population growth rates through"

Done.

33 - declines in what? growth rate?

We have added 'declines in population growth rates' to reflect the negative temperature sensitivity. L33.

53 - "Warmer environments are typically associated with smaller body size, higher mortality, faster growth, and earlier maturation (13-16)."

Done.

58 - This sentence feels out of place. I suggest deleting: "or anthropogenic effects. Consequently, life-history traits inform the reference points for fisheries species and likelihood of recovery for the overfished populations (20, 21)."

OK. Done.

75 - Delete this sentence to improve flow between paragraphs.

Done.

77 - what defines a population?

We have reworded 'populations' to 'population records'. L75, 77, and 424.

79 - "habitat temperature data are available from NOAA' World Atlas 2013"

Done.

85 - "Because the recent rate of change in the ocean temperatures in our study region has been minor (< 2 C per 100 years, 24, 26), we did not account for the small

temporal changes in the past sea temperatures." I don't understand this sentence and suggest it be deleted.

OK. Done.

100 - make all plural: "higher natural mortality rates (M), smaller asymptotic lengths (Ling), etc"

OK. Done.

123 - K, Linf, and M values (added Oxford comma)

Done.

159 - you called it a "life-table model" in the abstract. be consistent throughout.

Done.

175 - K, Ling, and model-derived A50 (added Oxford comma)

Done.

182 - reword as "but negative impacts on fish populations with slow life histories, especially elasmobranchs."

Done.

196 - "refuting in potential declines in population growth rate" - be very careful not to conflate population growth rate with population size - re-examine the MS to make sure this doesn't happen else where.

Done.

218 - suggest removing this text which feels tagged on "plasticity in spatial distribution is a mitigating factor to such temperature sensitivity." However, thanks for adding the previous text in response to my earlier comment.

OK. Done.

222 - "all but a few"

Done.

228 - "primarily focused"

Done.

231 - "population growth rates"

Done.

244 - The summary of this study remains incorrect. It measured the impact of historical warming on population growth rates (r , intrinsic growth rate) and found that some populations have been positively and negatively influenced by warming. This is similar to your finding that some life histories should benefit from warming while others should be negatively impacted. Different from your study which is entirely based on life history, its estimates are also subject to ecosystem effects (match-mismatches) and fisheries induced changes in age structure. The point is that "population sustainability" is incorrect - it measured impacts on "population growth rate" -- and it didn't contradict other models (the historical patterns differ from

forecast ones).

OK. We have revised this sentence about the study by Free et al. (2019) as suggested: “Given that recent research has found a mix of positive, negative, and neutral effects of historical climate warming on population growth rate (9)—differing from the forecast results of most existing models—more studies... are urgently needed.”. L252-257.

Figure 2 - I understand now why there are so many points at zero from your clear response. Can you add a sentence explaining this in the caption?

Done. We now add: “The vertical lines of data points at mean temperature (0 °C) reflect either a large number of single-population species or reef fishes with narrow ranges of habitat temperatures in our data.”

401 - This is incorrect. See comment and code above.

We have added 2 sentences to illustrate how rfishbase complements our data. See our response to major comment.

Reviewer #2 (Remarks to the Author):

The authors have dealt with most of my comments and the study should encourage future analyses of population growth responses. In any wide ranging comparative study like this there is always going to be a limitation on the accuracy of data but the authors have made sensible attempts to consider the robustness of their analyses.

Thank you.

The addition of the model fit with predictor BT into Table S6 (h-n) as I requested does appear to weaken the argument on Line 127 “of consistent signs of temperature effects on the mean responses of K, M, and for all 6 groups groups—positive slopes for K and M but negative slopes for Linf (Fig Fig. 3a–c; Supplemental Table S6)—even though some of these effects were not significantly different from zero, particularly for the pelagic and bathydemersal fishes (Fig. 3a–c)”. I therefore think some mention is made of this with respect to the relevant temperature measures for demersal and bathydemersal fish.

We have added a sentence about weaker effects of BT on life history traits compared to SST as suggested: “Also, bottom temperature exerted weaker effects on life history traits compared to SST, despite it is a relevant temperature measure to demersal or bathydemersal fishes (Table S6 h-n).” L129-131.

Line 20 . The authors may have misunderstood my point. I still don't think this sentence makes sense: Most marine fish species adapt to temperature increases by displaying higher mortality. Fish may be become subject to greater mortality if

predation and disease increases. The sentence on Line 53 does make sense as warmer environments are often associated with high mortality.

We have reworded this sentence into: “Most marine fish species express life history changes across a temperature gradient, such as faster growth, earlier maturation, and higher mortality at higher temperature.”. L20-21.